

# Impacts of bromine and iodine chemistry on tropospheric OH and HO₂: Comparing observations with box and global model perspectives

**Daniel Stone,[1] Tomás Sherwen,[2] Mathew J. Evans,[2,3] Stewart Vaughan,[1] Trevor Ingham,[1,4] Lisa K. Whalley,[1,4] Peter M. Edwards,[2] Katie A. Read,[2,3] James D. Lee,[2,3] Sarah J. Moller,[2,3] Lucy J. Carpenter, [2,3] Alastair C. Lewis, [2,3] Dwayne E. Heard[1,4]**

[1] School of Chemistry, University of Leeds, Leeds, UK
[2] Wolfson Atmospheric Chemistry Laboratories, Department of Chemistry, University of York, York, UK
[3] National Centre for Atmospheric Science, University of York, York, UK
[4] National Centre for Atmospheric Science, University of Leeds, Leeds, UK

## Abstract

The chemistry of the halogen species bromine and iodine has a range of impacts on tropospheric composition, and can affect oxidising capacity in a number of ways. However, recent studies disagree on the overall sign of the impacts of halogens on the oxidising capacity of the troposphere. We present simulations of OH and HO₂ radicals for comparison with observations made in the remote tropical ocean boundary layer during the Seasonal Oxidant Study at the Cape Verde Atmospheric Observatory in 2009. We use both a constrained box model, using detailed chemistry derived from the Master Chemical Mechanism (v3.2), and the three-dimensional global chemistry transport model GEOS-Chem. Both model approaches reproduce the diurnal trends in OH and HO₂. Absolute observed concentrations are well reproduced by the box model but are overpredicted by the global model, potentially owing to incomplete consideration of oceanic sourced radical sinks. The two models, however, differ in the impacts of halogen chemistry. In the box model, halogen chemistry acts to increase OH concentrations (by 9.8 % at midday at Cape Verde), while the global model exhibits a small increase in OH at Cape Verde (by 0.6 % at midday) but overall shows a decrease in the global annual mass weighted mean OH of 4.5 %. These differences reflect the variety of timescales through which the halogens impact the chemical system. On short timescales, photolysis of HOBr and HOI, produced by reactions of HO₂ with BrO and IO, respectively, increases the OH concentration. On longer timescales, halogen catalysed ozone destruction cycles lead to lower primary production of OH radicals through ozone photolysis, and thus to lower OH concentrations. The global model includes more of the longer timescale responses than the constrained box model and overall the global impact of the longer timescale response (reduced primary production due to lower O₃ concentrations) overwhelms the shorter timescale response (enhanced cycling from HO₂ to OH), and thus the global OH concentration decreases. The Earth system contains many such responses on a large range of timescales. This work highlights the care that needs to be taken to understand the full impact of any one process on the system as a whole.



## Introduction

Halogen chemistry in the troposphere influences budgets of $O_3$, $HO_x$ (OH and $HO_2$), $NO_x$ (NO and $NO_2$) (von Glasow et al., 2004; Saiz-Lopez and von Glasow, 2012; Simpson et al., 2015, Schmidt et al., 2016; Sherwen et al., 2016a; Sherwen et al., 2016b), affects the oxidation state of atmospheric mercury (Holmes et al., 2006; Holmes et al., 2010), and impacts aerosol formation (Hoffmann et al., 2001; O'Dowd et al., 2002; McFiggans et al., 2004; McFiggans et al., 2010; Mahajan et al., 2011; Sherwen et al., 2016c).

The production of bromine and iodine atoms in the marine boundary layer (MBL) following emissions of organohalogen compounds and the inorganic compounds $I_2$ and HOI has been shown to result in considerable destruction of tropospheric ozone (Read et al., 2008), leading to the production of bromine monoxide (BrO) and iodine monoxide (IO) radicals. Observations of BrO and IO radicals within the MBL have demonstrated widespread impacts on atmospheric composition and chemistry (Alicke et al., 1999; Sander et al., 2003; Leser et al., 2003; Saiz-Lopez and Plane, 2004; Saiz-Lopez et al., 2004; Peters et al., 2005; Saiz-Lopez et al., 2006; Whalley et al., 2007; Mahajan et al., 2010a; Commane et al., 2011; Dix et al., 2013; Gomez Martin et al., 2013), including significant effects on $HO_x$ concentrations and on the $HO_2$:OH ratio in coastal and marine locations (Bloss et al., 2005a; Sommariva et al., 2006; Bloss et al., 2007; Bloss et al., 2010; Kanaya et al., 2007; Whalley et al., 2010).

The role of halogens in in $HO_x$ chemistry was demonstrated during the NAMBLEX campaign in Mace Head, Ireland, (Heard et al., 2006), following several studies which attributed box model overestimates of $HO_2$ observations in marine environments to unmeasured halogen monoxides (Carslaw et al., 1999; Carslaw et al., 2002; Kanaya et al., 2001; Kanaya et al., 2002; Kanaya et al., 2007). Simultaneous measurements of OH and $HO_2$ by laser-induced fluorescence (LIF) (Bloss et al., 2005a; Smith et al., 2006) and halogen species by a combination of DOAS (for BrO or IO, OIO and $I_2$) (Saiz-Lopez et al., 2006) and broadband cavity ringdown spectroscopy (BBCRDS) (for OIO and $I_2$) (Bitter et al., 2005) during NAMBLEX enabled box model calculations to fully explore the impacts of halogens on the local composition. A box model without halogen chemistry was able to reproduce the NAMBLEX OH observations to within 25 %, but $HO_2$ observations were overestimated by up to a factor of 2 (Sommariva et al., 2006). The introduction of halogen chemistry, using DOAS measurements of BrO and IO (Saiz-Lopez et al., 2006) to constrain the model, increased the modelled OH concentrations by up to 15 % and decreased $HO_2$ by up to 30 % owing to reactions of $HO_2$ with XO radicals to form HOX which subsequently photolysed to X + OH (Sommariva et al., 2006). Bloss et al., (2005a) indicated that up to 40 % of the instantaneous $HO_2$ loss could be attributed to $HO_2$ + IO, and that photolysis of HOI was responsible for 15 % of the noontime OH production.



The impacts of halogen chemistry on $HO_x$ radicals at a site representative of the open ocean have been investigated
at the Cape Verde Atmospheric Observatory (CVAO). Measurements of halogen monoxides (Mahajan et al.,
2010a) at the site have been shown to have significant impacts on local ozone concentrations, notably in the
magnitude of the daily cycle (Read et al., 2008), and have been used to constrain box model calculations used to
explore observations of OH and $HO_2$ made during the RHaMBLe campaign in 2007 (Whalley et al., 2010). The
model calculations showed generally good comparisons with the observed OH and $HO_2$ concentrations, apart
from a period characterised by unusually high concentrations of HCHO. Compared to a model run in which
halogen chemistry was absent, bromine and iodine chemistry led to a 9 % increase in the modelled OH
concentration (Whalley et al., 2010). Owing to the dominance of the tropics in global methane oxidation (Bloss
et al., 2005b), such an impact of halogens on OH could have significant consequences for estimates of global
methane lifetimes, and on our understanding of the impacts of halogen chemistry on climate change.

In general, observationally constrained box model simulations suggest that halogens in the troposphere chemistry
will increase OH concentrations, primarily because of a change in the $HO_2$ to OH ratio occurring as a result of
reactions of halogen oxides (XO) with $HO_2$ to produce a hypohalous acid (HOX) which photolyses to give an OH
radical and a halogen atom (Kanaya et al., 2002; Bloss et al., 2005a; Kanaya et al., 2007; Sommariva et al., 2006;
Sommariva et al., 2007; Whalley et al., 2010). Other impacts on the $HO_x$ photochemical system are observed
(impacts from changes to $NO_x$ chemistry etc.) but these are minor and overall the general conclusion is that the
halogen chemistry tends to increase the OH concentration and thus the oxidising capacity of the atmosphere.

However, the observationally constrained studies are typically concerned with processes occurring at the surface,
and in a single location. The role of halogen chemistry in the troposphere as a whole is more uncertain, particularly
in the free troposphere and on a global scale (Saiz-Lopez and von Glasow, 2012; Simpson et al., 2015). Inclusion
of bromine chemistry in the three-dimensional (3D) chemistry transport model (CTM) MATCH-HPIC resulted
in decreases in tropospheric ozone concentrations of ~18 % over widespread areas, with regional decreases of up
to 40 % (von Glasow et al., 2004). Increases of more than 20 % were found for OH in the free troposphere, but,
globally, changes to OH were dominated by decreases in OH in the tropics owing to a reduction in primary
production from $O_3$ photolysis, leading to a decrease of 1-2 % in the global mean OH concentration (von Glasow
et al., 2004).

Significant decreases in tropospheric ozone (up to 30% at high latitude spring) were also reported for the
pTOMCAT model on inclusion of bromine chemistry (Yang et al., 2005). The CAM-Chem global chemistry-
climate model has shown an approximate 10 % decrease in global mean tropospheric ozone concentration on



incorporation of lower bromine emissions (Saiz-Lopez et al., 2012), while the GEOS-Chem CTM displays a
global decrease of 6.5 % (Parrella et al., 2012). The GEOS-Chem model indicated that bromine-catalysed loss
of ozone is limited by the rate of production of HOBr, and that $HO_2 + BrO$ is responsible for over 95 % of the
global tropospheric HOBr production. While HOBr can act as a source of OH on photolysis, the changes to $O_3$
and $NO_x$ resulting from the inclusion of bromine chemistry in GEOS-Chem led to a 4 % decrease overall in the
global annual mean OH (Parrella et al., 2012).

Vertically resolved airborne measurements of IO radicals in the free troposphere over the Pacific Ocean have also
demonstrated a role for iodine chemistry throughout the free troposphere, with IO observed at a mixing ratio of
~0.1 ppt in the free troposphere and found to be present in both recent deep convective outflow and aged free
tropospheric air (Dix et al., 2013). Model simulations to investigate iodine-driven ozone destruction throughout
the troposphere indicated that only 34 % of the total iodine-driven ozone loss occurs within the marine boundary
layer, with 40 % occurring in a transition layer and 26 % in the free troposphere (Dix et al., 2013).

The CAM-Chem and GEOS-Chem models have also been updated to encompass iodine chemistry, with results
from CAM-Chem showing iodine chemistry to be responsible for 17-27 % of the ozone loss in the tropical MBL
and 11-27 % of the ozone loss in the marine upper troposphere (Saiz-Lopez et al., 2014). The GEOS-Chem model
also showed iodine chemistry to be responsible for significant ozone destruction throughout the troposphere
(Sherwen et al., 2016a; Sherwen et al., 2016b; Sherwen et al., 2017). The GEOS-Chem simulations, which
incorporate chlorine, bromine and iodine chemistry, show a reduction in global tropospheric ozone concentration
of 18.6 %, compared to simulations with no halogen chemistry, a reduction in the global mean OH of 8.2 % to a
concentration of $1.28 \times 10^6$ $cm^{-3}$ and a resulting increase in global methane lifetime of 10.8 % to 8.28 years
(Sherwen et al., 2016b).

There is thus a discrepancy between box and global models as to the impact of halogen chemistry on OH
concentrations in the troposphere. Box models suggesting that OH radical concentrations should increase and
thus that halogens tend to increase the oxidising capacity, whereas the global models tend to suggest the opposite.

In this work, we use both a detailed chemical box model approach and a global chemistry-transport model to
investigate the local and global impacts of halogen chemistry on $HO_x$ radical concentrations. We focus on
seasonal $HO_x$ observations available from the Cape Verde Atmospheric Observatory (Vaughan et al., 2012). We
first provide a summary of the measurement site and the observations, followed by details of the two models used





in this study. We then evaluate the impact of halogens on the concentrations of oxidants in the two modelling
frameworks and consider the impact of halogen chemistry on global oxidising capacity.

## The Cape Verde Atmospheric Observatory

The Cape Verde Atmospheric Observatory is situated on the north east coast of the island of Sao Vicente (16.848
ºN, 24.871 ºW), approximately 500 km off the west coast of Africa. The observatory is in a region of high marine
biological production, and, for 95 % of the time, receives the prevailing northeasterly trade wind directly off the
ocean (Read et al., 2008; Carpenter et al., 2010). Measurements at the observatory are considered to be
representative of the open ocean, and CO, $O_3$, VOCs, $NO_x$ and $NO_y$ have been measured near-continuously at the
observatory since October 2006 (Lee et al., 2009; Carpenter et al., 2010).

In 2007, the observatory was host to the RHaMBLe intensive field campaign, during which a number of additional
measurements were made to complement the long-term measurements at the site (Lee et al., 2010), including LP-
DOAS measurements of halogen species (Read et al., 2008; Mahajan et al., 2010a) and formaldehyde (Mahajan
et al., 2010b), and LIF-FAGE measurements of OH and $HO_2$ (Whalley et al., 2010). The halogen monooxide
radicals BrO and IO exhibited a 'top-hat' diurnal cycle (Vogt et al., 1999; Vogt et al., 1996; Read et al., 2008;
Mahajan et al., 2010a) with essentially zero concentration in the hours of darkness and generally constant values
of approximately 2.5 ppt BrO and 1.4 ppt IO during the day.

The RHaMBLe campaign was followed by the Seasonal Oxidants Study (SOS) in 2009, during which
measurements of OH and $HO_2$ were conducted over three periods (Feb-March (SOS1), May-June (SOS2), and
September (SOS3)), and are discussed in detail by Vaughan et al. (2012). We present here the results from a
modelling study of the $HO_x$ measurements made during SOS1 and SOS2, when supporting measurements are
available, using both box and global model approaches. SOS3 is not considered in this work owing to a lack of
supporting measurements.

Measurements of OH and $HO_2$ during the Seasonal Oxidant Study were made by laser-induced fluorescence (LIF)
spectroscopy at low pressure using the fluorescence assay by gas expansion (FAGE) technique, and are described
in detail by Vaughan et al. (2012). Briefly, ambient air is drawn into a fluorescence cell situated on the roof of a
shipping container and maintained at pressures of ~ 2 Torr. The fluorescence cell has two excitation axes, with
excess NO added at the second axis to titrate $HO_2$ to OH, enabling simultaneous detection of OH and $HO_2$. OH
radicals in both excitation axes are excited by laser light at λ ~ 308 nm, generated by frequency tripling the output



of a solid state Nd:YAG pumped Ti:Sapphire laser system (Bloss et al., 2003). Channel photomultiplier tubes
coupled to gated photon counters are used to detect the $A^2\Sigma^+ - X^2\Pi_i$ OH fluorescence signal at $\lambda \sim 308$ nm.

Calibration of the instrument is achieved by measurement of the fluorescence signal from known concentrations
of OH and HO$_2$, produced by the photolysis of water vapour, and was performed over a range of conditions before,
during and after the campaign. For OH, the 1 $\sigma$ limit of detection (LOD) was in the range $(2\text{-}11) \times 10^5$ cm$^{-3}$ for a
5 min averaging period, while for HO$_2$ 1 $\sigma$ LOD was in the range $(6\text{-}13) \times 10^5$ cm$^{-3}$ for a 4 min averaging period.
Uncertainties (2 $\sigma$) in the measurements of OH and HO$_2$ are ~32 % (Vaughan et al., 2012).

Potential interferences in HO$_2$ measurements arising from conversion of alkene- and aromatic-derived peroxy
radicals to OH within the LIF detection cell, as described by Fuchs et al. (2011), are expected to be small for this
work owing to relatively low concentrations of alkenes and aromatics at the Cape Verde observatory (Carpenter
et al., 2010; Vaughan et al., 2012). Speciation of the peroxy radicals in the box model output (see Supplementary
Material) shows that 87.4 % of the peroxy radicals are HO$_2$ and CH$_3$O$_2$, 6.5 % CH$_3$C(O)O$_2$ and 1.1 % C$_2$H$_5$O$_2$,
all of which display no HO$_2$ interference in the laboratory (Whalley et al., 2013: Stone et al., 2014). Peroxy
radicals derived from OH-initiated oxidation of ethene and propene (HOC$_2$H$_4$O$_2$ and HOC$_3$H$_6$O$_2$, respectively)
were found to result in an interference signal for HO$_2$ in the laboratory (~40 % for the experimental configuration
in this work) but each radical comprises only ~0.6 % of the total RO$_2$ in this work. Thus, model calculations
reported here do not include representation of potential HO$_2$ interferences, although such phenomena may be
important in other environments (see for example, Whalley et al., 2013; Stone et al., 2014).

**Model Approaches**
We interpret the observations using two different modeling frameworks. The first is an observationally
constrained box model (DSMACC), the second is a global tropospheric chemistry transport model (GEOS-
Chem).

**Constrained Box Model**
The Dynamically Simple Model of Atmospheric Chemical Complexity (DSMACC) is described in detail by
Emmerson and Evans (2009) and Stone et al. (2010), and is a zero-dimensional model using the Kinetic Pre-
Processor (KPP) (Sandu and Sander, 2006). In this work we use a chemistry scheme based on a subsection of the
hydrocarbons (ethane, propane, *iso*-butane, *n*-butane, *iso*-pentane, *n*-pentane, hexane, ethene, propene, 1-butene,
acetylene, isoprene, toluene, benzene, methanol, acetone, acetaldehyde and DMS) available from the Master
Chemical Mechanism version 3.2 (MCM v3.2 http://mcm.leeds.ac.uk/MCM/home.htt) (Jenkin et al., 2003;





Saunders et al., 2003), with a halogen chemistry scheme described by Saiz-Lopez et al. (2006), Whalley et al.
(2010) and Edwards et al. (2011). We also include the reaction between OH and $CH_3O_2$ (Bossolasco et al., 2014;
Fittschen et al., 2014; Assaf et al., 2016; Yan et al., 2016), with a rate coefficient of $1.6 \times 10^{-10}$ cm$^3$ s$^{-1}$ (Assaf et
al., 2016) and products $HO_2$ + $CH_3O$ (Assaf et al., 2017), the impact of which on the $HO_2$:OH ratio and $CH_3O_2$
budget is described in the Supplementary Material. The total number of species in the model is ~1200, with ~5000
reactions. The full chemistry scheme used in the model is given in the Supplementary Data.

All measurements are merged onto a 10 minute timebase for input to the model and the model is run with
constraints applied as discussed in our previous work (Stone et al., 2010; Stone et al., 2011; Stone et al., 2014).
Concentrations of $CH_4$ and $H_2$ are kept constant at values of 1770 ppb (NOAA CMDL flask analysis,
[ftp://ftp.cmdl.noaa.gov/ccg/ch4/](ftp://ftp.cmdl.noaa.gov/ccg/ch4/)) and 550 ppb (Ehhalt and Rohrer, 2009; Novelli et al., 1999) respectively.
Formaldehyde measurements were not available during the SOS and we thus use HCHO concentrations generated
by the chemistry in the model, with the modelled HCHO concentrations in broad agreement with previous
measurements at the observatory (Mahajan et al., 2010b). Table 1 shows a summary of the input parameters to
the model.



| Species | Mean | Median | Range |
|---|---|---|---|
| $O_3$ / ppb | 33.8 ± 8.6 | 30.7 | 19.6 – 49.7 |
| CO / ppb | 102.3 ± 10.3 | 99.3 | 87.8 – 127.3 |
| $H_2O$ / ppm | 20542.3 ± 2753.8 | 21290.0 | 16778.5 – 24909.2 |
| NO / ppt | 11.2 ± 10.6 | 9.0 | 0.06 – 96.2 |
| Ethane / ppt | 961.3 ± 289.4 | 864.0 | 625.4 – 1799.2 |
| Propane / ppt | 136.1 ± 87.05 | 111.8 | 20.2 – 521.5 |
| *iso*-butane / ppt | 13.4 ± 9.8 | 11.1 | 0 – 62.7 |
| *n*-butane / ppt | 21.9 ± 17.6 | 17.8 | 0 – 112.9 |
| Actylene / ppt | 79.0 ± 27.8 | 70.4 | 45.0 – 180.5 |
| Isoprene / ppt | 0.1 ± 0.4 | 0 | 0 – 2.6 |
| *iso*-pentane / ppt | 3.9 ± 3.2 | 3.3 | 0 – 22.9 |
| *n*-pentane / ppt | 4.3 ± 3.0 | 3.9 | 0 – 21.7 |
| *n*-hexane / ppt | 1.0 ± 0.7 | 0.9 | 0 – 4.4 |
| Ethene / ppt | 43.6 ± 15.2 | 46.3 | 6.4 – 73.2 |
| Propene / ppt | 13.5 ± 3.6 | 13.0 | 6.2 - 24.1 |
| But-1-ene / ppt | 6.5 ± 1.4 | 6.3 | 3.5 – 10.6 |
| Benzene / ppt | 13.0 ± 17.0 | 8.3 | 0 – 64.4 |
| Toluene / ppt | 77.9 ± 388.8 | 0 | 0 – 2013.9 |
| Acetaldehyde / ppt | 511.8 ± 526.0 | 599.3 | 0 – 2622.6 |
| Methanol / ppt | 247.6 ± 336.2 | 173.3 | 0 – 3337.4 |
| DMS / ppt | 8.3 ± 38.3 | 0 | 0 – 291.8 |

Table 1: Summary of inputs to the model. Zero values indicate measurements below the limit of detection.
Further details can be found in Vaughan et al. (2012) and Carpenter et al. (2010).










Physical loss of each species in the model is represented by a first-order loss process, with the first-order rate
coefficient equivalent to a lifetime of approximately 24 hours, as discussed by Stone et al. (2010). Loss of reactive
species to aerosol surfaces is represented in the model by parameterisation of a first-order loss process to the
aerosol surface (Schwarz, 1986), as discussed by Stone et al. (2014).

A range of aerosol uptake coefficients for $HO_2$ have been reported in the literature, with recent measurements
indicating values of $\gamma_{HO2}$ between 0.003 and 0.02 on aqueous aerosols (George et al., 2013) while others have
reported values of $\gamma_{HO2} \sim 0.1$ (Taketani et al., 2008), and increased uptake coefficients in the presence of Cu and
Fe ions (Thornton et al., 2008; Mao et al., 2013).  In this work we use a value of $\gamma_{HO2} = 0.1$ in order to maintain
consistency with previous modelling studies at the site (Whalley et al., 2010) and to account for potential impacts
of ions of copper and iron in aerosol particles influenced by mineral dust (Carpenter et al., 2010; Muller et al.,
2010; Fomba et al., 2014; Matthews et al., 2014; Lakey et al., 2015).

The aerosol surface area in the model is constrained to previous measurements of dry aerosol surface area at the
observatory, corrected for differences in sampling height between the aerosol and $HO_x$ measurements and for
aerosol growth under humid conditions (Allan et al., 2009; Muller et al., 2010; Whalley et al., 2010).

Halogen species are constrained to a 'top-hat' profiles for BrO and IO (Vogt et al., 1999; Vogt et al., 1996; Read
et al., 2008), as observed during the RHAMBLE campaign in 2007 (Read et al., 2008; Mahajan et al., 2010a).
The observations indicate that while there is day to day variation in BrO and IO concentrations, there is little
seasonal variation (Mahajan et al., 2010a). BrO and IO are thus constrained to the mean observed mixing ratios
of 2.5 ppt and 1.4 ppt, respectively, for time points between 0930 and 1830 (GMT) and zero for all other times.
In a similar way to $NO_x$ (see Stone et al. (2010; 2011)), concentrations of all bromine or iodine species, including
BrO and IO, are permitted to vary according to the photochemistry as the model runs forwards.  At the end of
each 24 hour period in the model, the calculated concentrations of BrO and IO are compared to the constrained
value, and the concentrations of all bromine (Br, $Br_2$, BrO, HBr, HOBr, $BrONO_2$, $BrNO_2$, BrNO) and iodine (I,
$I_2$, IO, HI, HOI, INO, $INO_2$, $IONO_2$, OIO, $I_2O_2$, $I_2O_3$, $I_2O_4$, $HOIO_2$) species are fractionally increased or decreased
such that the calculated and constrained concentrations of BrO and IO are the same.  The model is run forwards
in time with diurnally varying photolysis rates until a diurnal steady state is reached, typically requiring between
5 and 10 days.





**Global Model**
We use the 3D global chemistry transport model GEOS-Chem (v10-01, www.geos-chem.org). The model has
been extensively evaluated against observations (Bey et al., 2001; Evans and Jacob, 2005; Nassar et al., 2009;
Mao et al., 2010; Zhang et al., 2010; Parrella et al., 2012; Hu et al., 2017). The model is driven by assimilated
winds calculated by the Goddard Earth Observing System at a horizontal resolution of $4° \times 5°$, with 47 vertical
levels from the surface to 50 hPa. Anthropogenic emissions of CO, $NO_x$ and $SO_2$ are described by the EDGAR
3.2 monthly global inventory (Olivier et al., 2005). Emissions of volatile organic compounds (VOCs) are
described by the RETRO monthly global inventory (van het Bolscher, 2008) for anthropogenic sources, with
ethane emissions described by Xiao et al., 2008, and the MEGAN v2.1 inventory (Guenther et al., 2006; Barkley
et al., 2011) for biogenic sources.

The $HO_x$-$NO_x$-VOC-$O_3$ chemistry scheme in the model is described in detail by Bey et al. (2001) and Mao et al.
(2013), with the isoprene oxidation mechanism described by Paulot et al. (2009). Photolysis rates use the FAST-
JX scheme (Bian and Prather, 2002; Mao et al., 2010), with acetone photolysis rates updated by Fischer et al.
(2012). Stratospheric chemistry is based on LINOZ McLinden et al. (2000) for $O_3$ and a linearised mechanism
for other species as described by Murray et al. (2012).

The model framework includes gas-aerosol partitioning of semi-volatile organic compounds (Liao et al., 2007;
Henze et al., 2007; Henze et al., 2009; Fu et al., 2008; Heald et al., 2011; Wang et al., 2011), and heterogeneous
chemistry (Jacob, 2000). Coupling between gas phase chemistry and sulfate-ammonium-nitrate aerosol is
described by Park et al. (2004) and Pye et al. (2009). A description of dust aerosol in the model is given by Fairlie
et al. (2007). Treatment of sea salt aerosol is described by Jaegle et al. (2011). The uptake coefficient for $N_2O_5$
uses the parameterisation by Evans and Jacob (2005), while that for $HO_2$ uses the parameterisation of Thornton
et al. (2008). A full description of the organic aerosol chemistry in the model is given by Heald et al. (2011).

The model includes recent updates to the chemistry scheme to include bromine chemistry (Parella et al., 2012;
Schmidt et al., 2016) and iodine chemistry (Sherwen et al., 2016a; Sherwen et al., 2016b). Sources of tropospheric
bromine in the model include emissions of $CHBr_3$, $CH_2Br_2$ and $CH_3Br$, and transport of reactive bromine from
the stratosphere. Debromination of sea-salt aerosol is not included in the model following the work of Schmidt et
al. (2016), which showed better agreement with observations of BrO made by the GOME-2 satellite (Theys et al.,
2011) and in the free troposphere and the tropical Eastern Pacific MBL (Gomez Martin et al., 2013; Volkamer et
al., 2015; Wang et al., 2015). Emission rates and bromine chemistry included in the model are described in detail
by Parella et al. (2012), with the additional bromine chemistry scheme described by 19 bimolecular reactions, 2



three-body reactions and 2 heterogeneous reactions using rate coefficients, heterogeneous reaction coefficients
and photolysis cross-sections recommended by Sander et al. (2011).

Iodine sources include emissions of $CH_3I$, $CH_2I_2$, $CH_2ICl$, $CH_2IBr$, $I_2$ and HOI. Emissions for $CH_3I$ follow Bell
et al. (2002), while those of other organic iodine species use parameterisations based on chlorophyll-a in the
Tropics and constant oceanic and coastal fluxes in extratropical regions (Ordonez et al., 2012). Emissions of
inorganic iodine species (HOI and $I_2$) use the results of Carpenter et al. (2013), with oceanic iodide concentrations
parameterised by MacDonald et al. (2014). The iodine chemistry scheme includes 26 unimolecular and
bimolecular reactions, 3 three-body reactions, 21 photolysis reactions and 7 heterogeneous reactions, using
recommendations by Atkinson et al. (2007) and Sander et al. (2011) where available. Full details are given by
Sherwen et al. (2016a; 2016b).

Photolysis rates of bromine and iodine compounds are calculated using the FAST-J radiative transfer model (Wild
et al., 2000; Bian and Prather, 2002; Mao et al., 2010). Wet and dry deposition are determined as for the standard
GEOS-Chem model (Liu et al., 2001; Wesely, 1989; Wang et al., 1998; Amos et al., 2012).

The tropospheric chemistry scheme is integrated using the SMVGEAR solver (Jacobson and Turco, 1994; Bey et
al., 2001).  The model, provides hourly output at the site of the Cape Verde Atmospheric Observatory. Model
simulations have been performed in the absence of halogens, with bromine chemistry, with iodine chemistry and
with bromine and iodine chemistry combined. Each model simulation is run for two years, with the analysis
performed on the second year (2009) and the first year discarded as model spin-up to enable evolution of long-
lived species.

**Model Results**
We now investigate the impact of halogen chemistry on tropospheric oxidation at Cape Verde within our two
modelling approaches.

**Constrained Box Model**
Figure 1 shows the observed and modelled time series for OH and $HO_2$ during SOS1 (February, March 2009) and
SOS2 (May, June 2009).  Observed concentrations of OH and $HO_2$ were typically higher in SOS2 than SOS1,
reaching maximum values in SOS2 of $\sim 9 \times 10^6$ cm$^{-3}$ OH and $4 \times 10^8$ cm$^{-3}$ $HO_2$ (Vaughan et al., 2012).  Similar
concentrations were observed in May and June 2007 during the RHaMBLe campaign (Whalley et al., 2010), with
a *t*-test indicating no statistically significant difference between the OH concentrations measured in May-June



2009 to those measured in May-June 2007 at the 95 % confidence level (Vaughan et al., 2012). Concentrations
of $HO_2$ measured in May-June 2009 were significantly higher than those measured in May-June 2007 at the 95
% confidence level, but were within the 1σ day-to-day variability (Vaughan et al., 2012). Temperatures during
SOS2 were typically higher than those during SOS1, with higher relative humidity during SOS2 (Vaughan et al.,
2012). Air masses during SOS1 had strong contributions from Atlantic marine air and African coastal region,
with polluted marine air and Saharan dust contributing ~ 20 % and 10 %, respectively, for the first half of the
measurement period. Conditions during SOS2 were typically cleaner, with Atlantic marine air representing the
major source, although coastal African air contributed ~ 40 % on some days. There was little influence from
polluted air, dust or continental air (Vaughan et al., 2012). Analysis of the variance of OH and $HO_2$ during SOS
indicated that ~70 % of the total variance could be explained by diurnal behaviour, with the remaining ~30 %
related to changes in air mass.

Figure 1 shows the observed and modelled time series for OH and $HO_2$, for model simulations with and without
halogen chemistry. For SOS1, the box model overpredicts the OH and $HO_2$ concentrations at the start of the
campaign (Julian days 59 and 61), but performs better for day 63, and captures both the observed diurnal profile
and the observed concentrations. For SOS2, the box model tends to agree better with the observations for both
OH and $HO_2$. A day-by-day comparison between the models and the observations is shown in the Supplementary
Material for days for which box model calculations were possible, which were limited by the availability of
supporting data.

Figure 2 shows the point-by-point model performance for OH and $HO_2$ for all data points combined, and for
SOS1 and SOS2 separately, for the full box model run including halogen chemistry. There is a tendency for
overprediction of OH and $HO_2$ during SOS1 (slopes of modelled vs observed concentrations are (1.86 ± 0.26) for
OH and (1.66 ± 0.21) for $HO_2$), which is dominated by the model overpredictions on days 59 and 61, with better
agreement observed during SOS2 (slopes of modelled vs observed concentrations are (1.11 ± 0.15) for OH and
(1.21 ± 0.12) for $HO_2$).

The measured and modelled average diurnal profiles of OH, $HO_2$ and the $HO_2$ to OH ratios are shown in Figure
3. At midday (1100-1300), the full model including halogen chemistry overpredicts OH by a median factor of
1.52 and $HO_2$ by a median factor of 1.21. A model run containing bromine chemistry but no iodine chemistry
gave median midday overpredictions of 1.40 and 1.30 for OH and $HO_2$, respectively, while a run containing
iodine but not bromine gave equivalent median overpredictions of 1.50 and 1.26, respectively. With no halogen
chemistry included in the model, the modelled OH decreases, giving a median overprediction at midday by a



factor of 1.37, while the modelled $HO_2$ increases, resulting in a median overprediction by a factor of 1.37 at
midday.

Thus the inclusion of halogens (bromine and iodine) in the box model changes the mean noon time OH and $HO_2$
concentrations by +9.8 % and –9.9 %, respectively. This impact of halogen chemistry is consistent in sign and
magnitude with previous studies (Kanaya et al., 2002; Bloss et al., 2005a; Kanaya et al., 2007; Sommariva et al.,
2006; Sommariva et al., 2007; Whalley et al., 2010).

Figure 4 shows the mean midday total $RO_x$ budget (given the fast processing time between $HO_2$ and HOBr/HOI
we identify the $RO_x$ family as OH, $HO_2$, HOBr, HOI, RO and $RO_2$) for the two measurement periods during SOS
for model runs with and without halogens.  These budgets are similar both between time periods (i.e. SOS1 vs
SOS2) and for box model calculations with and without halogen chemistry. Radical production dominated by
photolysis of ozone (~83 %), with photolysis of HCHO (~10 %), $CH_3CHO$ (~2 %) and $H_2O_2$ (~2 %) playing a
significantly smaller role. Radical termination reactions were dominated by $HO_2 + CH_3O_2$ (~23 %), aerosol uptake
of $HO_2$ (~21 %), $HO_2 + HO_2$ (~19 %), $CH_3C(O)O_2 + HO_2$ (~8 %), and $OH + HO_2$ (~5 %).  The inclusion of the
reaction between OH and $CH_3O_2$ reduces the importance of radical termination via $HO_2 + CH_3O_2$ (from ~26 %
of the total to ~23 % of the total), but otherwise has little impact on the total radical removal owing to the expected
production of $HO_2 + CH_3O$ (Assaf et al., 2017). Further details regarding the impact of the reaction between OH
and $CH_3O_2$ on the $HO_2$:OH ratio and $CH_3O_2$ budget are given in the Supplementary Material.

The budget analyses for SOS are consistent with those determined for the RHaMBLe campaign (Whalley et al.,
2010; Fittschen et al., 2014; Assaf et al., 2017), reflecting similarities in observed concentrations of long-lived
species and the method of the model constraint with observed $O_3$ concentrations and photolysis rates. The primary
source of radicals therefore remains fixed in all simulations, with the primary sinks for these species occurring
through radical-radical reactions. Thus, the total radical concentration and budget is little impacted by the
presence of halogens.

However, the partitioning of the radicals is impacted by the halogens. Without halogens the average midday
(1100-1300) $HO_2$ to OH ratio is (83.4 ± 15.4) (median = 82.7), with the halogens this changes to (68.3 ± 13.6)
(median = 66.9) (Table 2). This change in partitioning is mainly due to the reaction of $HO_2$ with BrO and IO
followed by the photolysis of HOBr and HOI to give OH. In this way the halogens tend to reduce the concentration
of $HO_2$ and increase the concentration of OH.





| | HO$_2$:OH ratio |
|---|---|
| Observed | 79.1 ± 34.1 (70.7) |
| | |
| Box model, no halogens | 83.4 ± 15.4 (82.7) |
| Box model with Br chemistry | 78.9 ± 15.6 (77.8) |
| Box model with I chemistry | 71.5 ± 13.0 (70.4) |
| Box model with Br and I chemistry | 68.3 ± 13.6 (66.9) |
| | |
| Global model, no halogens | 80.8 ± 18.1 (78.9) |
| Global model with Br chemistry | 81.9 ± 19.0 (79.7) |
| Global model with I chemistry | 70.4 ± 12.5 (70.5) |
| Global model with Br and I chemistry | 71.3 ± 13.2 (71.3) |


Table 2: Mean (± 1σ) midday (1100-1300 hours) ratios of HO$_2$ to OH (SOS1 and SOS2 combined). Median
values are given in parentheses.

In the box model without halogen chemistry, production of OH is dominated by ozone photolysis (76 %), HO$_2$ +
NO (12 %) and HO$_2$ + O$_3$ (9 %), with OH loss controlled by OH + CO (37 %), OH + CH$_4$ (16 %) and OH +
CH$_3$CHO (15 %), as shown in Figure 5. Production of HO$_2$ in the model excluding halogens is controlled by OH
+ CO (45 %), CH$_3$O + O$_2$ (19 %) and photolysis of HCHO (10 %), with loss governed by aerosol uptake (26 %),
HO$_2$ + HO$_2$ (26 %), HO$_2$ + NO (15 %), HO$_2$ + CH$_3$O$_2$ (12 %) and HO$_2$ + O$_3$ (10 %). In the presence of halogens,
the instantaneous budgets for OH and HO$_2$ are impacted by BrO and IO, as shown in Figures 5 and 6. For the
model run including halogens, OH production is still dominated by ozone photolysis (68 %), but there are
significant contributions from photolysis of HOI (10 %) and HOBr (3 %). Loss of HO$_2$ is also affected by the
presence of the halogen species, with the dominant loss processes including aerosol uptake (20 %), HO$_2$ + HO$_2$
(19 %), HO$_2$ + IO (14 %), HO$_2$ + NO (12 %), HO$_2$ + CH$_3$O$_2$ (11 %), HO$_2$ + O$_3$ (8 %) and HO$_2$ + BrO (6 %). As
shown in Figures 4-6 there is little difference in the radical budgets between SOS1 and SOS2.

This box modelling study is consistent with previous studies (Kanaya et al., 2002; Bloss et al., 2005a; Kanaya et
al., 2007; Sommariva et al., 2007; Whalley et al., 2010; Mahajan et al., 2010a; Stone et al., 2012) in that it implies
that halogen chemistry is likely to increase the OH concentration of the marine boundary layer (and potentially
other regions of the troposphere) as it enhances the HO$_2$ to OH conversion through the production of HOBr and
HOI. We now look at the impact of halogen chemistry on the concentrations of OH and HO$_2$ at Cape Verde within
the framework of a global atmospheric chemistry model.






**Global Model**

Figure 1 shows the time series for OH and $HO_2$ calculated by the global model GEOS-Chem, with the average diurnal profiles shown in Figure 3. The global model displays a significant overprediction for OH and $HO_2$ during SOS1, but exhibits reasonable skill at reproducing the observed concentrations during SOS2 and captures the $HO_2$:OH ratio for both measurement periods. The overpredictions of OH and $HO_2$ in the global model likely result from a combination of missing OH-sinks, particularly oxygenated volatile organic compounds (oVOCs) which are currently underestimated in the global model (Millet et al., 2015), and potential overprediction of the primary radical production rate owing to reductions in photolysis rates resulting from cloud cover that are not captured by the global model.

At midday (1100-1300), the modelled to observed ratios for OH and $HO_2$ for the global model excluding halogen chemistry are $(1.52 \pm 1.02)$ and $(1.72 \pm 0.80)$, respectively, with a mean modelled $HO_2$ to OH ratio of $(80.8 \pm 18.1)$ (compared to the observed $HO_2$ to OH ratio of $(79.1 \pm 34.1)$). For the global model run including bromine chemistry, but not iodine, the mean midday modelled to observed ratios for OH and $HO_2$ are $(1.48 \pm 1.05)$ and $(1.69 \pm 0.81)$, respectively, with a mean midday modelled $HO_2$ to OH ratio of $(81.9 \pm 19.0)$. Bromine chemistry thus acts to decrease the concentrations of both OH and $HO_2$, in contrast to the box model results which show increased concentrations of OH and decreased concentrations of $HO_2$. For the model run including iodine, but not bromine, the midday modelled to observed ratios for OH and $HO_2$ are $(1.57 \pm 1.00)$ and $(1.59 \pm 0.81)$, respectively, with a mean midday modelled $HO_2$ to OH ratio of $(70.4 \pm 12.5)$. Iodine chemistry thus results in increased OH and decreased $HO_2$ for both the global and box model simulations at Cape Verde. Inclusion of bromine and iodine chemistry combined leads to midday modelled to observed ratios of $(1.53 \pm 1.01)$ for OH and $(1.57 \pm 0.82)$ for $HO_2$, and a mean midday modelled $HO_2$ to OH ratio of $(71.3 \pm 13.2)$. These results are shown in Table 2, alongside those for the box model.

The results from the global model at Cape Verde thus differ from those of the box model. For the box model, inclusion of bromine and iodine chemistry, whether separately or combined, leads to increased OH and decreased $HO_2$ through increased conversion of $HO_2$ to OH through the production and subsequent photolysis of HOBr and HOI. In the global model a more complex pattern emerges. In a similar way to the box model, the $HO_2$ concentrations in the global model are decreased on inclusion of bromine and/or iodine owing to the additional loss reactions $HO_2$ + BrO and $HO_2$ + IO. When bromine chemistry is considered in the global model in isolation from iodine chemistry, the OH concentration decreases, despite the production and photolysis of HOBr. This decrease occurs as a result of a reduction in the $O_3$ concentration in the model on inclusion of bromine chemistry owing to the reaction of Br with $O_3$, which leads to a decrease in the rate of primary radical production from $O_3$




photolysis and thus lower OH concentrations. The impact of the decreased radical production rate is greater than
that leading to increased OH production through HOBr photolysis, and the net OH concentration is reduced in
the global model. This effect is not observed in the box model calculations as the model runs are constrained to
long-lived species – including $O_3$. The change in $O_3$ concentration on the inclusion of halogen chemistry is thus
not considered in the box model simulations, and the subsequent impacts of halogens consider only those changes
occurring on a more rapid timescale, which lead to increases in the OH concentration.

However, the inclusion of iodine chemistry in the global model does lead to increased OH concentrations at Cape
Verde. Direct emissions of HOI in the global model, in addition to chemical production through $HO_2 + IO$, result
in increased OH production through HOI photolysis as well as the repartitioning of $HO_2$ and OH through HOI
production in a similar manner to that for HOBr. However, the more rapid cycling of $HO_2$ to OH through the
more rapid production and photolysis of HOI compared to HOBr, reduces the impact of iodine chemistry on the
$HO_2$:OH ratio compared to that for bromine chemistry. Iodine chemistry thus can reduce the OH concentration
similarly to bromine chemistry, through the destruction of $O_3$ and subsequent reduction in primary production
rate, but the impact is less than that for bromine, and can be offset by the direct emissions of HOI which increases
the production rate of OH through photolysis.

The impacts of iodine chemistry in the global model are thus more complex than those for bromine chemistry.
When bromine and iodine chemistry are combined in the global model there is a competition between the effects
of the reduction in primary production of OH, through depletion of $O_3$, and the production of OH from photolysis
of HOBr and HOI and, for the model simulations at Cape Verde, the impacts of direct HOI emissions dominate.
The OH concentration is thus marginally increased compared to simulations containing no halogens, although the
$HO_2$ concentrations are significantly decreased.

The impacts of halogens on OH radical concentrations in the global model thus display a complexity that is
somewhat obscured in the box model simulations. Overall, the inclusion of halogens in the global model leads to
a slight increase in OH at Cape Verde, but, owing to the opposing effects of bromine and iodine, this result is
subject to the modelled concentrations of bromine and iodine species. Observations at Cape Verde made between
November 2006 and June 2007 indicate 'top-hat' profiles for BrO and IO, with average daytime mixing ratios of
2.5 ppt and 1.4 ppt, respectively, and little variability over the entire campaign (Read et al., 2008; Mahajan et al.,
2010a). The global model simulations reported here predict average mixing ratios of ~0.5 ppt for BrO and ~1 ppt
for IO during SOS, and thus underpredict BrO but perform well for IO. The underprediction of BrO at Cape Verde
results from recent model updates which exclude emissions of bromine species from sea-salt debromination



(Schmidt et al., 2016) in order to provide improved agreement with observations of BrO made by the GOME-2
satellite (Theys et al., 2011) and in the free troposphere and the tropical Eastern Pacific MBL (Gomez Martin et
al., 2013; Volkamer et al., 2015; Wang et al., 2015). We now discuss the global impacts of halogen chemistry.

**Global impacts of halogen chemistry on OH and HO$_2$**
On the global scale, concentrations of OH and HO$_2$ are reduced on inclusion of bromine and iodine chemistry,
both individually and combined. The global mass weighted annual mean OH concentration decreases by 3.8 %
on inclusion of bromine chemistry, but only 0.02 % on inclusion of iodine chemistry alone. When the chemistry
of bromine and iodine is combined in the model, the global mass weighted annual mean OH concentration
decreases by 4.5 %. For HO$_2$, the global mass weighted annual mean is decreased by 4.2 % by bromine, 5.6 % by
iodine and 9.7 % by bromine and iodine combined. Figure 7 shows the probability distribution functions for the
changes to OH and HO$_2$ concentrations for the monthly mean values for all grid boxes within the troposphere for
the year. For the majority of grid boxes, concentrations of OH and HO$_2$ are reduced on inclusion of bromine
chemistry, with iodine also generally reducing HO$_2$ concentrations but leading to a wider spread of changes to
the OH concentration, and similar numbers of grid boxes showing increased and decreased concentrations. When
bromine and iodine chemistry are combined in the model, HO$_2$ shows a more significant decrease than for either
halogen individually, and OH, although exhibiting increased concentrations in a significant number of grid boxes,
displays a greater tendency for decreased concentrations.

Figure 8 shows the changes to the annual modelled surface layer OH and HO$_2$ concentrations on inclusion of
halogen chemistry, with annual surface layer mixing ratios of BrO and IO shown in Figure 9. The most significant
changes to OH and HO$_2$ occur over marine regions, particularly over the Southern Pacific. Smaller changes are
observed over land, and any increased concentrations, including those for OH over Cape Verde, can be seen to
occur in coastal regions where the impacts of direct HOI emissions dominate and concentrations of IO
concentrations are typically elevated.

Thus, overall, halogens act to reduce the oxidising capacity of the troposphere through reductions to O$_3$ and
subsequent reductions in the primary production rates of OH and HO$_2$, despite the slight increase in OH
concentration predicted by the global model at Cape Verde. Consideration of the full extent of the impacts of
halogens on the global oxidising capacity is hindered by uncertainties in the concentrations and distributions of
halogen species, and model representations of halogen processes, particularly those relating to sea salt
debromination, ocean iodide emissions, parameterisations of iodine recycling in aerosols and photolysis of higher
iodine oxides (Sherwen et al., 2016a).




## Conclusions

Measurements of OH and HO$_2$ made by LIF-FAGE at the Cape Verde Atmospheric Observatory during the Seasonal Oxidants Study in 2009 have been simulated by a constrained box model and a three-dimensional global chemistry transport model. The observations are generally reproduced well by the box model, but are overpredicted by the global model.

The oxidising capacity of the two models, as manifested by the OH concentration, shows opposing sensitivity to halogens. The constrained box model shows an increase in OH concentrations with the inclusion of halogens, whereas the global transport model shows a decrease in OH concentrations globally, despite a marginal decrease in the OH concentration at Cape Verde. This difference between models reflects differing representation of chemical timescales by the models. The box model is constrained to concentrations of long-lived compounds, including O$_3$, and considers only impacts on short timescales, whereas the global model includes impacts occurring on longer timescales. Within this context, the box model includes the short timescale impact of halogens on the repartitioning of HO$_2$ to OH, thus increasing OH and decreasing HO$_2$, but does not consider the longer timescale impact of halogen-mediated ozone destruction which impacts primary radical production. This highlights a general problem with understanding the complex interactions within atmospheric chemistry and the Earth system in general. Evaluating the impact of a small part of the system on the system as a whole can be difficult and the most significant processes may occur on timescales significantly longer than those of the perturbation.

## Acknowledgements

This project was funded by the Natural Environment Research Council (NERC, NE/E011403/1), with support of the Cape Verde Atmospheric Observatory by the National Centre for Atmospheric Science (NCAS) and the SOLAS project. DS would also like to thank NERC for the award of an Independent Research Fellowship (NE/L010798/1). Computational resources were provided by the NERC BACCHUS project (NE/L01291X/1)

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



**Figures**

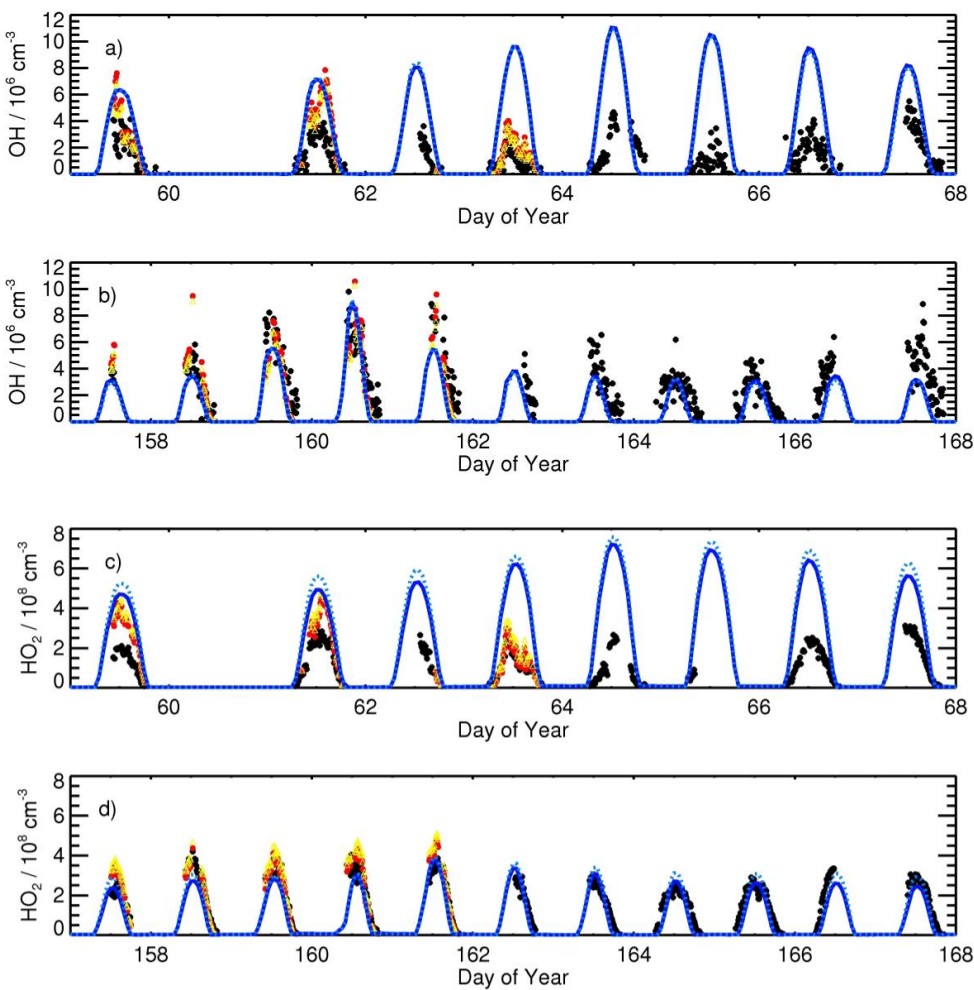



Figure 1: Observed and modelled concentrations of a) OH during SOS1 (February-March 2009, days 58-68); b)
OH during SOS2 (May-June 2009, days 157-168); c) $HO_2$ during SOS1; d) $HO_2$ during SOS2. Observed data are
shown in black; box model concentrations with halogen chemistry are shown by filled red circles; box model
concentrations without halogen chemistry are shown by open orange triangles; global model concentrations with
halogen chemistry are shown by solid dark blue lines; global model concentrations without halogen chemistry
are shown by broken blue lines.





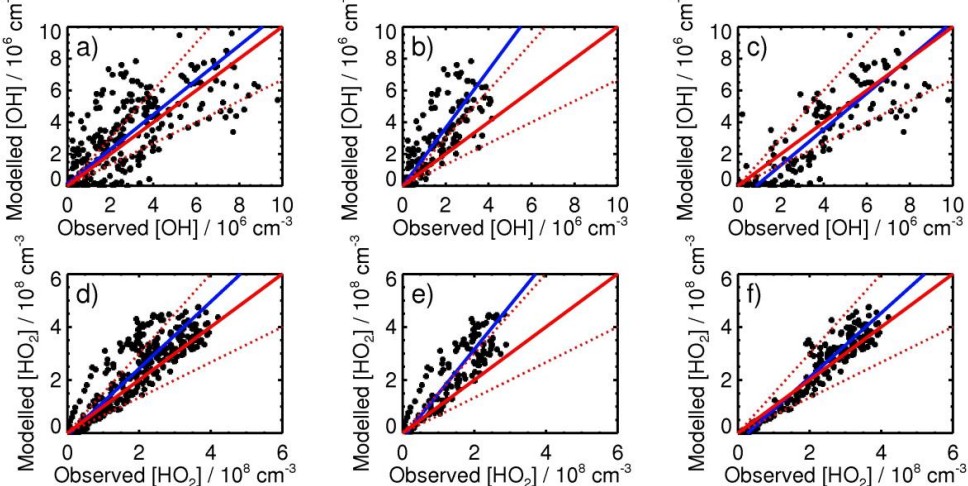


Figure 2: Comparison of modelled and observed concentrations of a) OH during SOS1 (February-March 2009)
and SOS2 (May-June 2009); b) OH during SOS1; c) OH during SOS2; d) $HO_2$ during SOS1 and SOS2; e) $HO_2$
during SOS1; f) $HO_2$ during SOS2. In each plot, the solid red line indicates the 1:1 line, with 50 % limits given
by the broken red lines. The best fit lines are shown in blue and are described by a) $[OH]_{mod} = (1.09 \pm 0.11) \times$
$[OH]_{obs} + (0.13 \pm 0.38) \times 10^6$ ($r^2 = 0.49$); b) $[OH]_{mod} = (1.82 \pm 0.26) \times [OH]_{obs} - (0.01 \pm 0.51) \times 10^6$ ($r^2 = 0.56$);
c) $[OH]_{mod} = (1.11 \pm 0.15) \times [OH]_{obs} - (0.95 \pm 0.66) \times 10^6$ ($r^2 = 0.64$); d) $[HO_2]_{mod} = (1.26 \pm 0.10) \times [HO_2]_{obs} -$
$(0.08 \pm 0.22) \times 10^8$ ($r^2 = 0.77$); e) $[HO_2]_{mod} = (1.66 \pm 0.21) \times [HO_2]_{obs} - (0.17 \pm 0.34) \times 10^8$ ($r^2 = 0.78$); f) $[HO_2]_{mod}$
$= (1.21 \pm 0.12) \times [HO_2]_{obs} - (0.32 \pm 0.30) \times 10^8$ ($r^2 = 0.91$).



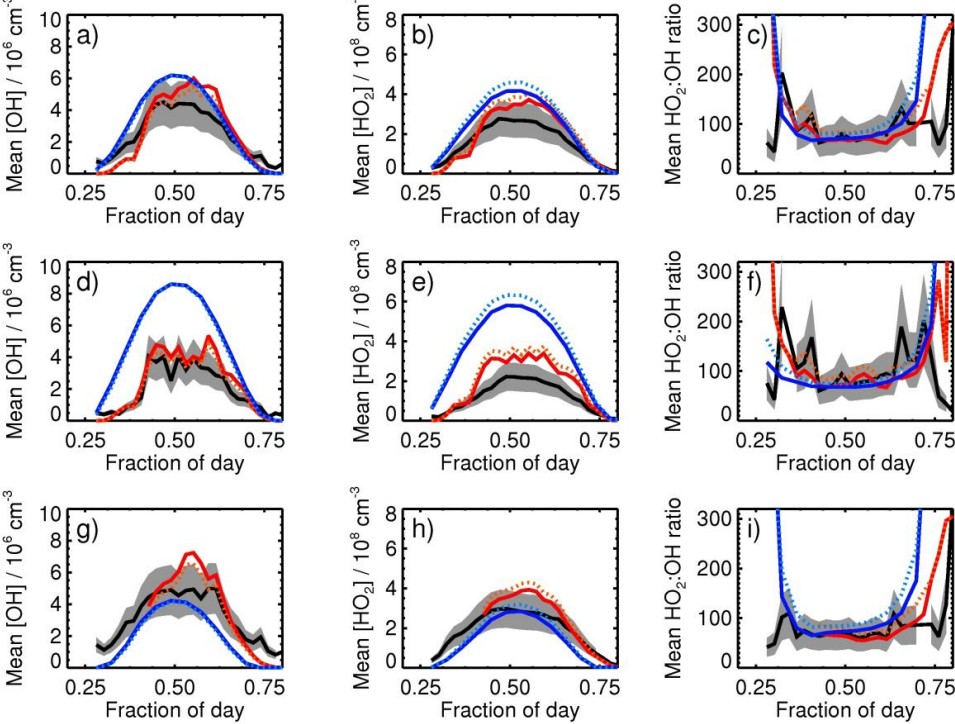


Figure 3: Average diurnal profiles during the Seasonal Oxidant Study (SOS) at the Cape Verde Atmospheric
Observatory for a) OH during both measurement periods; b) $HO_2$ during both measurement periods; c) $HO_2$:OH
ratio during both measurement periods; d) OH during SOS1 (Feb-Mar 2009); e) $HO_2$ during SOS1; f) $HO_2$:OH
during SOS1; g) OH during SOS2 (May-June); h) $HO_2$ during SOS2; i) $HO_2$:OH ratio during SOS2. Observed
data are shown in black, with grey shading indicating the variability in the observations; box model output with
halogen chemistry is shown by solid red lines; box model output without halogen chemistry is shown by broken
orange lines; global model output with halogen chemistry is shown by solid dark blue lines; global model output
without halogen chemistry is shown by broken blue lines.



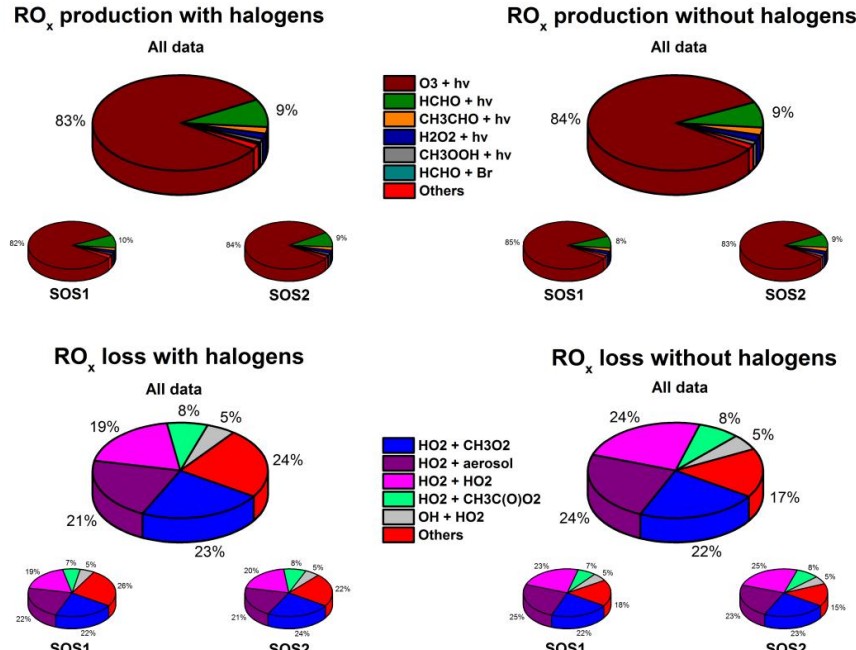


Figure 4: Processes controlling the a) instantaneous $RO_x$ radical production (with $RO_x$ defined here as $OH+HO_2+HOBr+HOI+RO+RO_2$ owing to the rapid processing between $HO_2$ and $HOBr/HOI$) around noon (1100-1300 hours) for box model simulations with halogen chemistry; b) the instantaneous $RO_x$ radical production around noon for box model simulations without halogen chemistry; c) the instantaneous $RO_x$ radical loss around noon for box model simulations with halogen chemistry; d) the instantaneous $RO_x$ radical loss around noon for box model simulations without halogen chemistry. The main charts show the average results for SOS1+SOS2, with results for SOS1 and SOS2 shown separately in the inset charts.





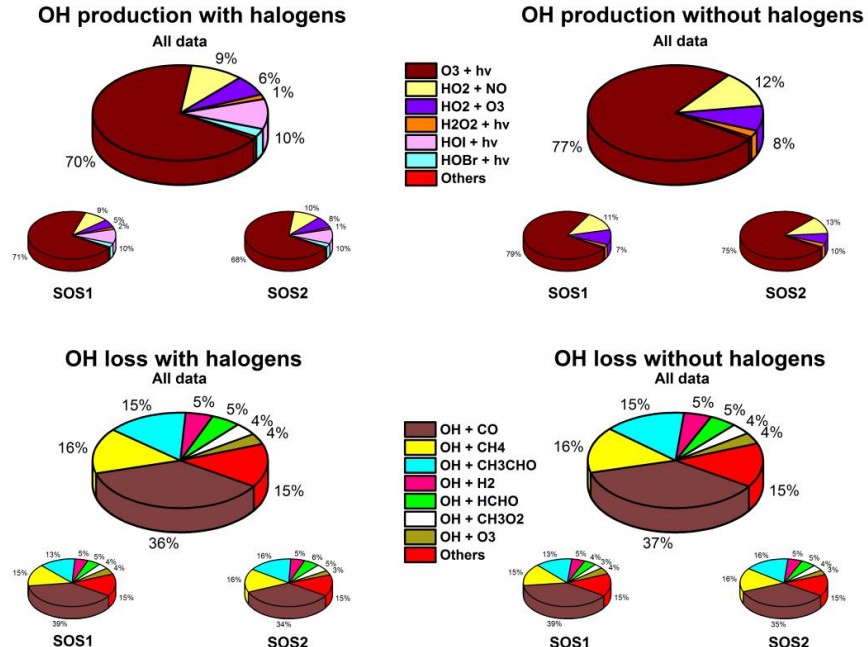

086

Figure 5: Processes controlling the a) instantaneous OH radical production around noon (1100-1300 hours) for box model simulations with halogen chemistry; b) the instantaneous OH radical production around noon for box model simulations without halogen chemistry; c) the instantaneous OH radical loss around noon for box model simulations with halogen chemistry; d) the instantaneous OH radical loss around noon for box model simulations without halogen chemistry. The main charts show the average results for SOS1+SOS2, with results for SOS1 and SOS2 shown separately in the inset charts.




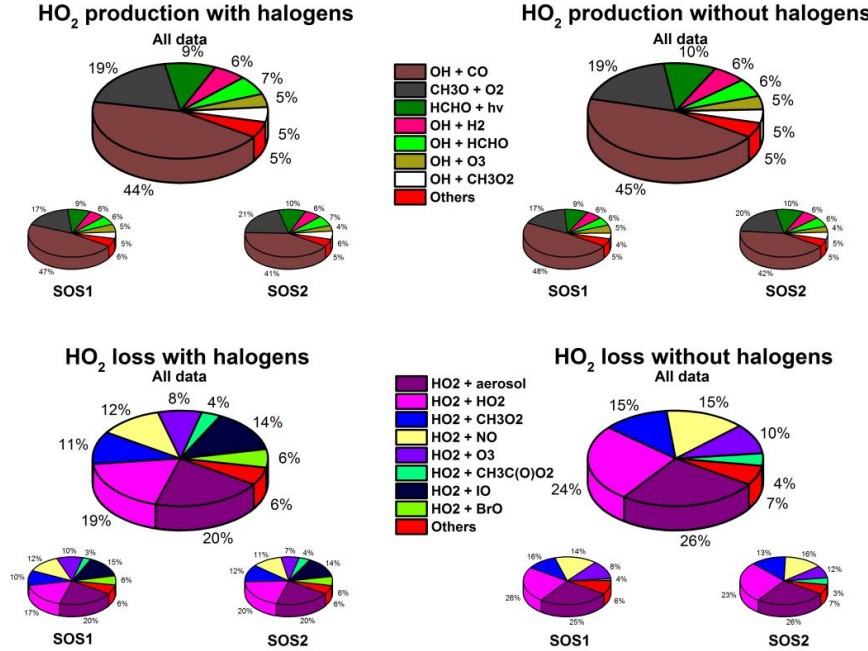


Figure 6: Processes controlling the a) instantaneous HO$_2$ radical production around noon (1100-1300 hours) for
box model simulations with halogen chemistry; b) the instantaneous HO$_2$ radical production around noon for box
model simulations without halogen chemistry; c) the instantaneous HO$_2$ radical loss around noon for box model
simulations with halogen chemistry; d) the instantaneous HO$_2$ radical loss around noon for box model simulations
without halogen chemistry.  The main charts show the average results for SOS1+SOS2, with results for SOS1
and SOS2 shown separately in the inset charts.







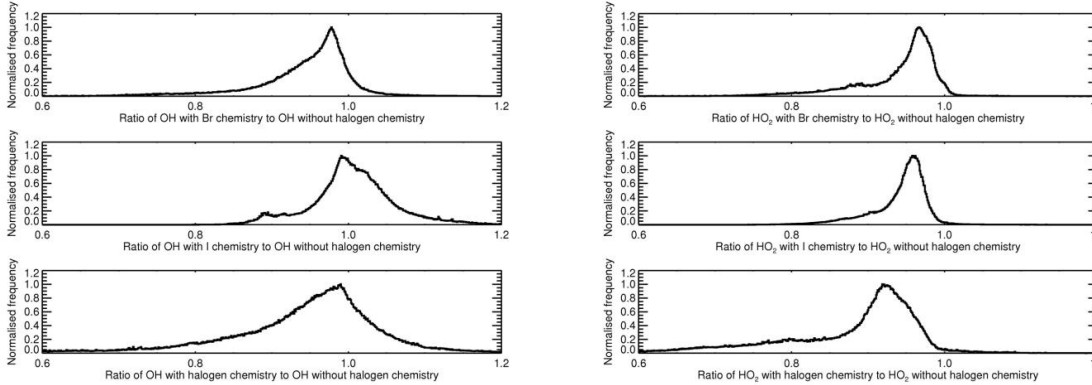


Figure 7: Normalised probability distribution functions showing the fractional changes in OH (left hand side) and
HO$_2$ (right hand side) in GEOS-Chem for all grid boxes on inclusion of bromine chemistry (upper panels), iodine
chemistry (middle panels) and bromine and iodine chemistry combined (lower panels).






Figure 8: Percentage changes to annual surface layer concentrations of OH (left hand side) and HO₂ (right hand
side) in GEOS-Chem on inclusion of bromine chemistry (upper panels), iodine chemistry (middle panels) and
bromine and iodine chemistry combined (lower panels).





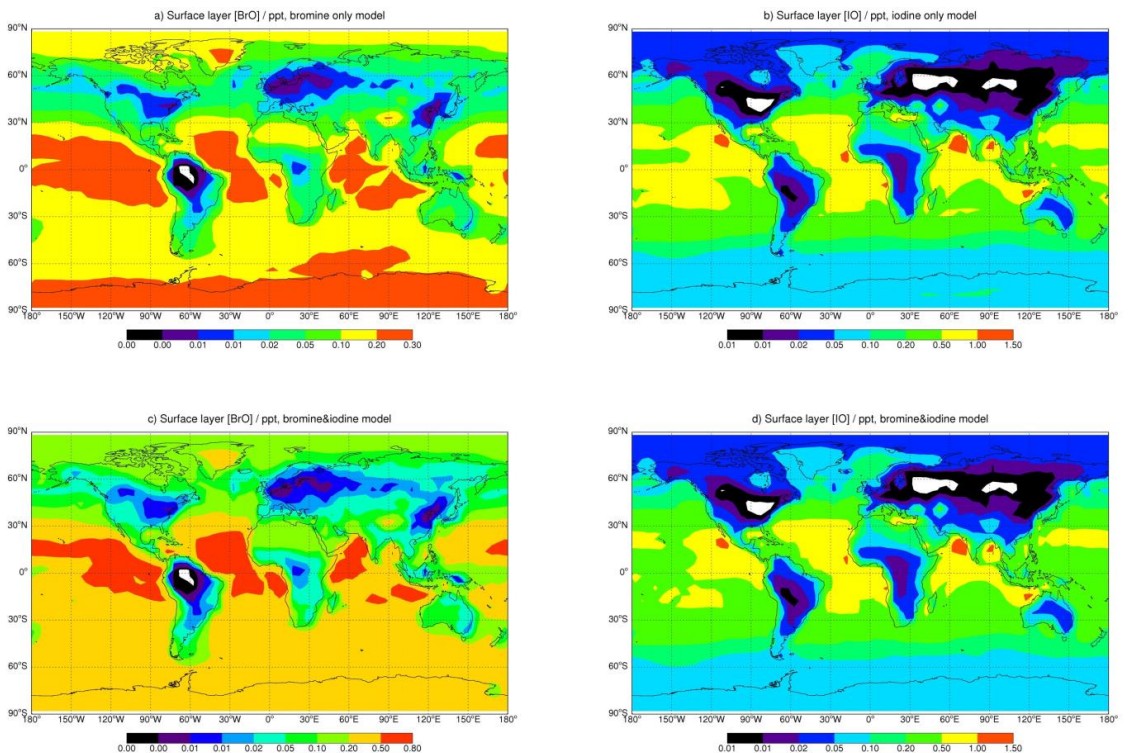


Figure 9: Annual surface layer mixing ratios (ppt) of BrO and IO radicals in GEOS-Chem for model runs with just bromine chemistry (upper left panel), just iodine chemistry (upper right panel) and bromine and iodine chemistry combined (lower panels).


