# Peer review of "Impacts of bromine and iodine chemistry on tropospheric OH and HO2"

_Atmospheric Chemistry and Physics, 2017_

## Referee Comment (RC1) · Anonymous Referee #1 · 17 Nov 2017

Stone et al. presents a study on the influence of tropospheric halogens on OH and HO2. The study is based on observations from the Cape Verde Atmospheric Observatory, a chemistry box model, and a global chemical transport model. Overall, the paper is well written and well suited for ACP. I recommend it being accepted following minor revision. I list specific comments and questions below:

1) Line 49: Le Breton et al (2017, doi: 10.1016/j.atmosenv.2017.02.003) have also presented measurements of BrO in the MBL, consider including a citation.

2) Line 224: What are the processes behind "Physical loss", is it deposition? If so, I suggest you state this explicitly. Is a lifetime of 24 hour reasonable?

[Figure]

3) Line 244 to 253: The assumption of constant BrO and IO between 9:30 and 18:30 seems overly crude. As can be seen in Figure 3 of Read et al (reference given in manuscript), it takes approximately 6 hours for BrO and IO to raise from 0 to max. How would the results of the box model study change is more realistic assumptions about the diurnal cycle of BrO and IO are used?

4) Line 288 to 291: Please specify the additional bromine reactions that were added to the Parella mechanism.

5) Line 477-481: Consider extending the discussion of sea-salt debromination. Schmidt et al (reference given in manuscript) presented two simulations: one without sea salt debromination and one including it. The simulation that included sea salt debromination led to levels of BrO that appeared biased high compared to satellite observations, but reproduce the levels observed in the mid Atlantic MBL (see Figure S8 of Schmidt et al.). How would the result of the global model change is sea salt debromination was included? Also, consider commenting on the resent study by Chen et al (2017, doi: 10.1002/2017GL073812) that show that sulfur chemistry may provide a missing sink of Bry in the MBL to balance the sea salt debromination source.
* * *

---

## Referee Comment (RC2) · Anonymous Referee #2 · 26 Dec 2017

The paper presents an interesting comparison between observations of OH and HO2 at the Cape Verde Observatory and two modelling approaches to evaluate how bromine and iodine chemistry impacts HOx. The results obtained by the two different model setups are discussed and evaluated in the context of the different chemical schemes and timescales considered in the two models. I found the paper interesting and well structured, presenting results in a clear and complete format and hence I recommend the paper for publication in ACP. Please find below two questions and minor comments for consideration by the authors: - In page 9, line 244, it is written that the box model is constrained by the mean observed mixing ratio of BrO and IO, 2.5 and 1.4 pptv, respectively. What was the peak value of BrO and IO in the model runs?, if the peak

value used was 2.5 and 1.4, did you run a sensitivity with the XO peak values measured by Read et al., 2008 and Mahajan et al., 2010 ? - One thing that I miss in this paper is the how halogen-driven changes in NOx affect HOx?. There is not mention to this aspect and from previous modelling studies it is expected to have a bearing on HOx. I would suggest the authors to mention whether they have explored this coupling and perhaps add some additional results to the revised manuscript.

Minor Comments: P2,L54: " in in HOx..", please remove one "in". P3,L81: please replace "troposphere" by tropospheric ? P31,Fig.1: For clarity, please improve quality in the box model results, e.g. what represents the yellow color?

---

## Author Response (AR1)

We would like to thank the reviewers for their comments which have improved the manuscript. Specific
comments from the reviewers are given in bold below, followed by our responses in plain text.
*Referee #1*
**Stone et al. presents a study on the influence of tropospheric halogens on OH and HO$_2$. The study is based on**
**observations from the Cape Verde Atmospheric Observatory, a chemistry box model, and a global chemical**
**transport model. Overall, the paper is well written and well suited for ACP. I recommend it being accepted**
**following minor revision. I list specific comments and questions below:**
**1) Line 49: Le Breton et al (2017, doi: 10.1016/j.atmosenv.2017.02.003) have also presented measurements of**
**BrO in the MBL, consider including a citation.**
We have included the citation.
**2) Line 224: What are the processes behind "Physical loss", is it deposition? If so, I suggest you state this**
**explicitly. Is a lifetime of 24 hour reasonable?**
The physical losses incorporate losses such as wet and dry deposition and diffusion out of the box for any model
generated species with a sufficiently long lifetime. In our previous work (Stone et al., 2010; Stone et al., 2014)
we investigated the sensitivity to the deposition rate, with results indicating that variation of the deposition
lifetime between 1 hour and 5 days results in little change to the modelled concentrations of OH and HO$_2$. We
have added the following details to the manuscript for clarification:
*Deposition processes, including dry deposition and wet deposition, and diffusion are represented in the model by*
*a first-order loss process, with the first-order rate coefficient equivalent to a lifetime of approximately 24 hours.*
*As discussed by Stone et al. (2010), variation of the deposition lifetime between 1 hour and 5 days results in*
*limited changes to the modelled concentrations of OH and HO$_2$.*
**3) Line 244 to 253: The assumption of constant BrO and IO between 9:30 and 18:30 seems overly crude. As**
**can be seen in Figure 3 of Read et al. (reference given in manuscript), it takes approximately 6 hours for BrO**
**and IO to raise from 0 to max. How would the results of the box model study change is more realistic**
**assumptions about the diurnal cycle of BrO and IO are used?**
The peak daytime concentrations of BrO and IO are typically reached between 9 am and 10 am, with increases
from zero occurring from approximately 6 am, as shown in Figure 3 of Read et al. The discussion in the
manuscript is unaffected by the early morning concentrations of BrO and IO owing to the focus on the midday
(11 am to 1 pm) concentrations of OH and HO$_2$, and the HO$_2$:OH ratio, which are influenced on short timescales.
The data presented in Figures 1-3 are potentially affected by the assumption of zero concentrations of BrO and
IO before 9:30 am, and we have performed simulations in which concentrations of BrO and IO are set to increase
from 6 am until 10 am, at which point the constant concentrations are reached, using the data shown in Figure 3
of Read et al. However, the low concentrations of BrO and IO during this period, and the relatively few data
points for OH and HO$_2$ during these times (there were no measurements of OH or HO$_2$ at these times of day during SOS2, as shown in Figures 1 and 3), the data presented in the manuscript are not significantly affected.
This is shown in the figure below (Figure S6), combined with the response to referee 2. We propose to combine
this figure and to provide the details in the supplementary information to the manuscript.
*4) Line 288 to 291: Please specify the additional bromine reactions that were added to the Parella mechanism.*
We apologise for the confusion here. The additional bromine reactions are those described by Parella et al. (i.e.
in addition to the standard mechanism which does not include any halogen chemistry). We have removed the
reference to "additional" in order to clarify this:
*Emission rates and bromine chemistry included in the model are described in detail by Parella et al. (2012), with*
*the bromine chemistry scheme described by 19 bimolecular reactions, 2 three-body reactions and 2*
*heterogeneous reactions using rate coefficients, heterogeneous reaction coefficients and photolysis cross-sections*
*recommended by Sander et al. (2011).*
*5) Line 477-481: Consider extending the discussion of sea-salt debromination. Schmidt et al (reference given*
*in manuscript) presented two simulations: one without sea salt debromination and one including it. The*
*simulation that included sea salt debromination led to levels of BrO that appeared biased high compared to*
*satellite observations, but reproduce the levels observed in the mid Atlantic MBL (see Figure S8 of Schmidt et*
*al.). How would the result of the global model change is sea salt debromination was included?*
We have expanded the discussion of this in the manuscript to include the following details:
*The global model simulations reported here predict average mixing ratios of ~0.5 ppt for BrO and ~1 ppt for IO*
*during SOS, and thus underpredict BrO but perform well for IO. The underprediction of BrO at Cape Verde*
*results from recent model updates which exclude emissions of bromine species from sea-salt debromination*
*(Schmidt et al., 2016) in order to provide improved agreement with observations of BrO made by the GOME-2*
*satellite (Theys et al., 2011) and in the free troposphere and the tropical Eastern Pacific MBL (Gomez Martin et*
*al., 2013; Volkamer et al., 2015; Wang et al., 2015). If sea-salt debromination were included, daytime mixing*
*ratios of BrO at Cape Verde would be approximately 2 ppt, as shown by Parella et al. (2012) and Schmidt et al.*
*(2016), and thus in closer agreement to the observations. Increased modelled concentrations of BrO at Cape*
*Verde resulting from inclusion of sea-salt debromination would have a greater effect on OH and $HO_2$, leading to*
*more significant decreases in OH and $HO_2$ when bromine chemistry is included without iodine chemistry, with*
*the larger decrease in OH potentially off-setting the increase in OH observed when bromine and iodine chemistry*
*are combined. However, the current model simulations do not consider the coupling between bromine and sulfur*
*chemistry, which may represent a significant sink for reactive bromine species in the troposphere and balance*
*sources from sea-salt debromination (Chen et al., 2017). These results thus demonstrate the need for further*
*investigation and constraint of sources and emission rates of bromine species, and of the coupling between sulfur*
*chemistry and reactive bromine species. We now discuss the global impacts of halogen chemistry.*

*Also, consider commenting on the resent study by Chen et al (2017, doi: 10.1002/2017GL073812) that show*
*that sulfur chemistry may provide a missing sink of Bry in the MBL to balance the sea salt debromination*
*source.*
We have now included details of this study in our discussion of sea-salt debromination (see above).

*Referee #2*

*The paper presents an interesting comparison between observations of OH and HO$_2$ at the Cape Verde*
*Observatory and two modelling approaches to evaluate how bromine and iodine chemistry impacts HO$_x$. The*
*results obtained by the two different model setups are discussed and evaluated in the context of the different*
*chemical schemes and timescales considered in the two models. I found the paper interesting and well*
*structured, presenting results in a clear and complete format and hence I recommend the paper for publication*
*in ACP. Please find below two questions and minor comments for consideration by the authors:*
*- In page 9, line 244, it is written that the box model is constrained by the mean observed mixing ratio of BrO*
*and IO, 2.5 and 1.4 pptv, respectively. What was the peak value of BrO and IO in the model runs?, if the peak*
*value used was 2.5 and 1.4, did you run a sensitivity with the XO peak values measured by Read et al., 2008*
*and Mahajan et al., 2010?*

The peak values used in the model simulations were 2.5 ppt BrO and 1.4 ppt IO. We have now performed a
sensitivity analysis to these mixing ratios, using the upper and lower limits to the diurnal averages reported by
Read et al. (3.5 ppt and 2.0 ppt for BrO and 2.0 ppt and 1.0 ppt for IO). The impacts of these changes on the
diurnal profiles for OH and HO$_2$, and on the HO$_2$:OH ratio, are shown in the figure below (Figure S6). Given the
focus of the manuscript on the different trends between the box and global models, rather than the absolute
concentrations of BrO and IO, we propose to include this figure (Figure S6), which also includes sensitivity to
early morning concentrations of BrO and IO as suggested by reviewer 1, in the supplementary information to the
paper for completeness. We have added a comment in the main text (line 248) to refer to the sensitivity to BrO
and IO concentrations ("*Sensitivity to these mixing ratios is discussed in the Supplementary Material*") and have
added the following details to the Supplementary Material:

*Observations of BrO and IO at the Cape Verde Atmospheric Observatory show average diurnal mixing ratios of*
*2.5 ppt and 1.4 ppt, respectively, which were used as constraints in the box model simulations presented in this*
*work. In order to test the sensitivity of OH and HO$_2$ to these constraints, we have performed simulations in which*
*the BrO and IO mixing ratios were constrained to the upper and lower limits of the observed values (3.5 ppt and*
*1.5 ppt for BrO and 2.0 ppt and 1.0 ppt for IO, as reported by Read et al. (2008) and Mahajan et al. (2010)). In*
*addition, simulations were performed in which mixing ratios of BrO and IO increase from 0600 hours and reach*
*the constant values of 2.5 ppt and 1.4 ppt, respectively, as shown by Read et al. (2008). Results from these*
*simulations are shown in Figure S6, and indicate that there are only minor differences in the OH and HO$_2$*
*concentrations between simulations performed on constraining to the upper and lower limits of the observed BrO*
*and IO concentrations, and that there is little sensitivity of OH or HO$_2$ to the early morning mixing ratios of BrO*
*and IO.*

[Figure]

*Figure S6: Impacts of changes to BrO and IO constraints on average diurnal profiles for a) OH during both*
*measurement periods; b) HO$_2$ during both measurement periods; c) HO$_2$:OH ratio during both measurement*
*periods; d) OH during SOS1 (Feb-Mar 2009); e) HO$_2$ during SOS1; f) HO$_2$:OH during SOS1; g) OH during*
*SOS2 (May-June); h) HO$_2$ during SOS2; i) HO$_2$:OH ratio during SOS2. Observed data are shown in black, with*
*grey shading indicating the variability in the observations; model simulations constrained to the average daytime*
*(0930 to 1830 hours) mixing ratios of BrO (2.5 ppt) and IO (1.4 ppt) are shown in red; simulations constrained*
*to the upper limits to the daytime mixing ratios of BrO (3.5 ppt) and IO (2.0 ppt) are shown in dark blue;*
*simulations constrained to the lower limits to the daytime mixing ratios of BrO (1.5 ppt) and IO (1.0 ppt) are*
*shown in light blue; simulations constrained to the average daytime mixing ratios of BrO (2.5 ppt) and IO (1.4*
*ppt) and including increases in the mixing ratios from 0600 hours are shown by the dashed purple lines;*
*simulations with no halogens are shown by the dashed orange lines.*

*- One thing that I miss in this paper is the how halogen-driven changes in NO$_x$ affect HO$_x$?. There is not*
*mention to this aspect and from previous modelling studies it is expected to have a bearing on HO$_x$. I would*
*suggest the authors to mention whether they have explored this coupling and perhaps add some additional*
*results to the revised manuscript.*

While we do not explicitly discuss the impacts of halogen-driven changes in NO$_x$ on HO$_x$, this is included in the
model simulations and there are some details in the discussion of the radical budgets and in Figures 4-6 which
show how the role of NO$_x$ chemistry in controlling radical budgets changes on inclusion of halogen chemistry.
We propose to include the following details to discuss these impacts more thoroughly:

*The change in the relative importance of $HO_2 + NO$ on inclusion of halogens in the model results from both the*
*increase in the total $HO_2$ sink, owing to the additional losses through $HO_2 + BrO$ and $HO_2 + IO$, and the shift in*
*$NO_x$ partitioning owing to the reactions $Br + NO \rightarrow Br + NO_2$ and $IO + NO \rightarrow I + NO_2$. The reactions of BrO*
*and IO with NO result in a change in the $NO_2$:NO ratio of approximately 10 %, on average, which reduces the*
*impact of $HO_2 + NO$ as both a sink for $HO_2$ and a source for OH.*
**Minor Comments: P2,L54: " in in $HO_x$..", please remove one "in".**
We have corrected this.
**P3,L81: please replace "troposphere" by tropospheric?**
We have changed this sentence to:
*In general, observationally constrained box model simulations suggest that halogens in the troposphere will*
*increase OH concentrations.*
**P31,Fig.1: For clarity, please improve quality in the box model results, e.g. what represents the yellow color?**
The yellow data points represent the box model concentrations including halogen chemistry. We have corrected
the figure caption, which previously stated the points to be orange, and have made some improvements to the
presentation of the figure (symbol size, scale) to improve the clarity. Additional plots also are provided in the
supplementary material to show these data in more detail

[revised manuscript text omitted]

**Supplementary Material**

**Speciation of modelled peroxy radicals**

Figure S1 shows the speciation of peroxy radicals during SOS determined by the box model. The dominant species at Cape Verde are HO$_2$ and CH$_3$O$_2$, which comprise 87.4 % of the total peroxy radical concentration, and are followed by CH$_3$C(O)O$_2$ (6.5 %) and C$_2$H$_5$O$_2$ (1.1 %), all of which display no HO$_2$ interference in the laboratory (Whalley et al., 2013; Stone et al., 2014). Any peroxy species potentially contributing to interferences in HO$_2$ measurements thus constitutes < 4 % of the total peroxy radical concentration, with each species representing < 1 % of the total. Potential interferences arising from conversion of alkene- and aromatic-derived peroxy radicals to OH within the LIF detection cell, as described by Fuchs et al. (2011), are thus expected to be small for SOS and are not explicitly described in the model for this work.

[Figure]

Figure S1: Speciation of peroxy radicals during SOS1 and SOS2. Radical names are as given by the MCM.

**Impact of CH$_3$O$_2$ + OH**

Recent experiments have indicated a rapid reaction between CH$_3$O$_2$ and OH (Bossolasco et al., 2014; Fittschen et al., 2014; Assaf et al., 2016; Yan et al., 2016), with the dominant products expected to be CH$_3$O + HO$_2$ at an observed yield of $(0.8 \pm 0.2)$ (Assaf et al., 2017). As shown in Figures 5 and 6 (main text), this reaction contributes 4 %, on average, to the total midday OH loss during SOS and 5 % to the total HO$_2$ production, assuming 100 % yield of CH$_3$O + HO$_2$. Inclusion of the reaction in the chemistry scheme, for model runs in which halogens are included, decreases the modelled concentration of OH at midday from $5.3 \times 10^6$ cm$^{-3}$ to $5.2 \times 10^6$ cm$^{-3}$, and increases the HO$_2$ concentration from $3.2 \times 10^8$ cm$^{-3}$ to $3.9 \times 10^8$ cm$^{-3}$.

Figure S2 shows the mean modelled diurnal profile for CH$_3$O$_2$ during SOS, for model runs with and without the reaction between CH$_3$O$_2$ and OH. Inclusion of the reaction decreases the mean midday CH$_3$O$_2$ concentration by 24 %, from $5.7 \times 10^8$ cm$^{-3}$ to $4.6 \times 10^8$ cm$^{-3}$, and thus has a more significant impact on CH$_3$O$_2$ than on OH or HO$_2$. Similar changes to modelled OH, HO$_2$ and CH$_3$O$_2$ were reported by Assaf et al. (2017) using an updated MCM based model for the RHaMBLe campaign in Cape Verde in 2007 (Whalley et al., 2010).

[Figure]

Figure S2: Mean box modelled diurnal profiles for a) OH; b) HO$_2$; c) CH$_3$O$_2$ during SOS (SOS1 and SOS2 combined) for model runs with (red solid line) and without (red broken line) the reaction between CH$_3$O$_2$ and OH. For OH and HO$_2$, observations are shown in black, with grey shading indicating the variability in the observations.

Figure S3 shows the mean midday (1100 to 1300 hours) budgets for CH$_3$O$_2$ during SOS for model runs with and without the reaction between CH$_3$O$_2$ and OH. Midday production of CH$_3$O$_2$, both with and without the reaction between CH$_3$O$_2$ and OH, is dominated by CH$_3$ radical production from CH$_4$ + OH (~ 55 %), and is followed by the reactions of CH$_3$C(O)O$_2$ radicals with NO (~ 17 %) and other RO$_2$ radicals (~ 15 %). Midday loss of CH$_3$O$_2$, when CH$_3$O$_2$ is included, is dominated by reactions with NO (~ 41 %), HO$_2$ (~ 33 %), the reaction with OH (~ 15 %), and CH$_3$O$_2$ self-reaction (~ 8 %). If the reaction of CH$_3$O$_2$ with OH is not included in the model, the loss reaction with NO represents ~ 48 % of the total $CH_3O_2$ loss and the reactions with $HO_2$ and other $CH_3O_2$ radicals represent ~ 36 % and ~ 12 %, respectively.

[Figure]

Figure S3: Mean midday (1100 to 1300 hours) budgets for $CH_3O_2$ during SOS (SOS1 and SOS2 combined). Panel a) production of $CH_3O_2$ for model runs with the reaction between $CH_3O_2$ and OH; b) production of $CH_3O_2$ for model runs without the reaction between $CH_3O_2$ and OH; c) loss of $CH_3O_2$ for model runs with the reaction between $CH_3O_2$ and OH; d) loss of $CH_3O_2$ for model runs without the reaction between $CH_3O_2$ and OH.

**Time series for observed and box modelled OH and HO$_2$ radical concentrations**

Figures S4 and S5 shows the times series for OH and HO$_2$ observations and model simulations day-by-day during

SOS1 and SOS2.

[Figure]

Figure S4: Times series for observed and modelled OH during SOS1 (days 59 to 63) and SOS2 (days 157 to 162).

Observed data are shown in black; box model concentrations with halogen chemistry are shown by filled red circles; box model concentrations without halogen chemistry are shown by open yellow triangles.

[Figure]

Figure S5: Times series for observed and modelled HO$_2$ during SOS1 (days 59 to 63) and SOS2 (days 157 to

162). Observed data are shown in black; box model concentrations with halogen chemistry are shown by filled red circles; box model concentrations without halogen chemistry are shown by open yellow triangles.

**417 Box model sensitivity of OH and HO$_2$ to BrO and IO mixing ratios**

Observations of BrO and IO at the Cape Verde Atmospheric Observatory show average diurnal mixing ratios of

2.5 ppt and 1.4 ppt, respectively, which were used as constraints in the box model simulations presented in this work. In order to test the sensitivity of OH and HO$_2$ to these constraints, we have performed simulations in which the BrO and IO mixing ratios were constrained to the upper and lower limits of the observed values (3.5 ppt and

1.5 ppt for BrO and 2.0 ppt and 1.0 ppt for IO, as reported by Read et al. (2008) and Mahajan et al. (2010)). In addition, simulations were performed in which mixing ratios of BrO and IO increase from 0600 hours and reach the constant values of 2.5 ppt and 1.4 ppt, respectively, as shown by Read et al. (2008). Results from these simulations are shown in Figure S6, and indicate that there are only minor differences in the OH and HO$_2$

concentrations between simulations performed on constraining to the upper and lower limits of the observed BrO

and IO concentrations, and that there is little sensitivity of OH or $HO_2$ to the early morning mixing ratios of BrO

and IO.

[Figure]

Figure S6: Impacts of changes to BrO and IO constraints on average diurnal profiles for a) OH during both
measurement periods; b) $HO_2$ during both measurement periods; c) $HO_2$:OH ratio during both measurement
periods; d) OH during SOS1 (Feb-Mar 2009); e) $HO_2$ during SOS1; f) $HO_2$:OH during SOS1; g) OH during
SOS2 (May-June); h) $HO_2$ during SOS2; i) $HO_2$:OH ratio during SOS2. Observed data are shown in black, with
grey shading indicating the variability in the observations; model simulations constrained to the average
daytime (0930 to 1830 hours) mixing ratios of BrO (2.5 ppt) and IO (1.4 ppt) are shown in red; simulations
constrained to the upper limits to the daytime mixing ratios of BrO (3.5 ppt) and IO (2.0 ppt) are shown in dark
blue; simulations constrained to the lower limits to the daytime mixing ratios of BrO (1.5 ppt) and IO (1.0 ppt)
are shown in light blue; simulations constrained to the average daytime mixing ratios of BrO (2.5 ppt) and IO
(1.4 ppt) and including increases in the mixing ratios from 0600 hours are shown by the dashed purple lines;
simulations with no halogens are shown by the dashed orange lines.